# Multi-scale mean field learning for adaptive decision making in multi-agent systems

Guowen Li[1,2], Jiaxin Du[1,2], Zhenzhen Zhao[1,2], Guojiang Shen[1,2] and Xiangjie Kong[1,2]

[1] College of Computer Science and Technology, Zhejiang University of Technology, Hangzhou, China
[2] Zhejiang Key Laboratory of Visual Information Intelligent Processing, Hangzhou, China



## ABSTRACT

In multi-agent reinforcement learning (MARL), the large number of agents can lead to information overload, hindering effective learning in large-scale systems such as industrial control and smart manufacturing environments. While mean field methods offer scalable solutions in homogeneous agent environments, real-world scenarios often involve heterogeneous agent decision making and dynamic interaction structures. To address these challenges, attention-based adaptive mean field methods have emerged. However, they still face limitations: (1) neighbor-weighted mean field lacks a global perspective, limiting the ability to model global coordination in systems; (2) single-scale mean field representations struggle to capture multi-level agent interactions critical for scene interaction optimization. To overcome these limitations, we propose a multi-scale mean field reinforcement learning framework, integrating far-field global action distributions with near-field local interactions weighted by attention. By leveraging multi-head attention, our method comprehensively captures interactions at different scales, enabling more adaptive decision making. Experimental results demonstrate superior performance and scalability across various benchmark tasks, highlighting its potential for enhancing decision-making in complex environments.

## INTRODUCTION

In recent years, multi-agent reinforcement learning (MARL) has been widely applied across various domains, including traffic scheduling (*Shen et al., 2024b*; *Yang et al., 2023*), robotic collaboration (*Zhu, Dastani & Wang, 2024*; *Park et al., 2023*), unmanned aerial vehicle (UAV) assisted (*Shen et al., 2024a*; *Kong et al., 2024*), and resource allocation (*Gebrekidan, Stein & Norman, 2024*; *Li et al., 2023*; *Wang et al., 2024a*), enabling pervasive intelligent technology integration. However, with technological advancements, a key challenge faced by MARL is the rapid growth in the number of agents. In practical applications, as the agent population expands, traditional MARL methods often encounter issues such as high computational complexity, low decision making efficiency, and insufficient accuracy, hindering their deployment in large-scale environments. Therefore,

Corresponding author
Xiangjie Kong, xjkong@ieee.org

developing scalable and efficient MARL methods has become crucial for advancing automation, decision making, and intelligence in diverse sectors.

A key challenge faced by MARL is the rapid growth in the number of agents M. In practical applications, this growth leads to an explosive increase in the dimensionality of the joint action space ($A = \prod_{i=1}^{M} A_i$) and the state/observation information relevant for interaction modeling, causing issues such as high computational complexity, low decision making efficiency, and insufficient accuracy. To address this problem, researchers initially proposed mean field reinforcement learning (MFRL) (*Ren et al., 2023*; *Chen et al., 2024*). This approach abstracts interactions among agents by approximating them as interactions between an agent and a virtual mean agent, thereby reducing computational complexity. However, by constructing a "completely balanced" mean field, MFRL struggles to capture complex, real-world agent interactions, especially in large-scale and complex environments where decision making often depends on dynamic and heterogeneous agent interactions. Recent advancements in adaptive mean field methods have employed attention mechanisms to assign higher weights to neighboring agents, better modeling local interactions. However, these methods face two key limitations: (1) Lack of multi-level interaction modeling: Existing approaches rely on a single-layer mean field, which cannot comprehensively capture hierarchical interactions in large-scale systems. (2) Limited global perspective: The focus on local agent interactions neglects the influence of global agent distributions, which is critical for optimizing complex system-level decision making.

To address the challenges outlined above, we propose a multi-scale mean field reinforcement learning method tailored for large-scale, heterogeneous systems. Specifically, existing mean field reinforcement learning methods struggle to simultaneously capture both local interactions among neighboring agents and the global influences of the entire system. To overcome this limitation, we introduce the multi-scale mean field representation, which integrates near-field (local agent interactions) and far-field (global system effects) through a dual-scale learning framework. This approach enhances the model's adaptability to the heterogeneity of local environments and improves policy refinement, allowing for more accurate decision making across different levels of interaction. By overcoming the oversimplification of traditional mean field methods, our approach is well-suited to scale to large systems, offering a robust solution for complex, multi-agent scenarios. We further provide the mathematical formulations underpinning this method, ensuring both its implementability and theoretical grounding.

Additionally, we propose the Multi-Scale Mean Field Q-learning (MSMF-Q) reinforcement learning method, an improved version of the traditional Mean Field Q-Learning (MF-Q) approach. The MSMF-Q method integrates a multi-head attention mechanism to dynamically assign weights to near-field neighbor information, while combining it with far-field data to capture multi-level interaction features. This mechanism allows the model to precisely capture the details of agent interactions across different scales, thereby simulating realistic agent behaviors in complex environments. While all agents in our experiments share the same observation and action space

(*i.e.*, structurally homogeneous), the proposed attention mechanism allows the model to capture heterogeneous influences from different neighbors. This interaction-level heterogeneity enables the agent to focus on more relevant or critical neighbors, thereby improving decision-making in complex systems. Extensive experiments across various scenarios validate the effectiveness of our proposed method, demonstrating its scalability and performance in large-scale, multi-agent systems.

The main contributions of this work are as follows:

- We propose a novel multi-scale mean field representation to address the limitations of traditional single-scale methods that fail to capture agent interactions at multiple levels. Furthermore, we provide the formal mathematical framework for the multi-scale mean field representation, ensuring its theoretical and operational foundation.
- We introduce MSMF-Q, a multi-scale reinforcement learning approach. By incorporating an attention mechanism into the global mean field, our method dynamically integrates local mean fields, enabling joint learning of both scales.
- We conduct extensive experiments to validate the effectiveness of the proposed methods across various scenarios. The experimental results demonstrate that the multi-scale mean field representation outperforms existing approaches in terms of scalability, performance, and adaptability, proving its applicability in large-scale, multi-agent environments.

The structure of this article is as follows: In 'Related Work', we summarize the latest advancements in MARL and mean field methods. 'Preliminary' primarily covers the prior knowledge and relevant formulas involved in the proposed algorithm. 'Methodology' begins by introducing the framework of the proposed method, followed by a modular breakdown of the implementation details of the framework. 'Experiments' describes our experimental setup and provides a brief analysis of the experimental results. Finally, 'Conclusion' concludes the article.

## RELATED WORK

### Multi-agent reinforcement learning

MARL methods have been widely applied in various scenarios, giving rise to diverse algorithms tackling different challenges (*Zhang et al., 2021*; *Hegde & Bouroche, 2024*; *Oroojlooy & Hajinezhad, 2023*). Independent Proximal Policy Optimization (IPPO) (*De Witt et al., 2020*) trains each agent separately using Proximal Policy Optimization (PPO), offering stable policy optimization *via* effective exploration-exploitation trade-offs. Multi-Agent Proximal Policy Optimization (MAPPO) (*Yu et al., 2022*), in contrast, adopts centralized optimization to improve overall coordination. Heterogeneous-Agent Proximal Policy Optimization (HAPPO) (*Kuba et al., 2022*) further enhances communication efficiency and accelerates convergence under the centralized training and execution paradigm.

DGN (*Jiang et al., 2018*) promotes cooperation by learning latent features through graph convolution and applying relational regularization. Multi-Agent Deep Deterministic

Policy Gradient (MADDPG) (*Lowe et al., 2017*) enables agents to explore latent action spaces more effectively using random perturbations, improving adaptability in complex environments. Finally, Nash Q-learning (*Yang et al., 2018*) introduces game-theoretic equilibrium optimization to MARL, and serves as a foundational component in many mean-field-based algorithms.

Although the aforementioned MARL algorithms show high performance in practical applications, their interaction logic limits their applicability. These algorithms are typically more suited for scenarios with fewer or simpler interactions and struggle to handle the challenges of large-scale, complex interaction environments.

## Mean field multi-agent reinforcement learning

To simplify interaction modeling in multi-agent systems, the mean field (MF) method was introduced (*Jusup et al., 2023*; *Li et al., 2024*; *Wu et al., 2024*), where the influence of other agents is approximated by a virtual average agent. Building on this idea, MF-Q (*Yang et al., 2018*) applied Nash Q-learning to mean field settings and demonstrated superior performance over value-based approaches like MF-AC. Subsequent works introduced parameterized distributions to capture population-level behavior more accurately (*Perrin et al., 2021*, *2022*), though these methods generally assume homogeneous agents and struggle with complex heterogeneous interactions.

To address such limitations, approaches like multi-typed mean fields (*Ganapathi Subramanian et al., 2020*) have been proposed for handling heterogeneous agent groups. General Mean Field Q-Learning (GMF-Q) (*Guo et al., 2019*) improves policy stability by integrating action distributions and stochastic policies using a Boltzmann exploration strategy. In the context of density regulation, *Cui & Li (2024)* apply robust mean field learning with stochastic disturbance modeling to enhance control over large-scale homogeneous robotic swarms.

Although these mean field based MARL methods effectively address the training complexity in systems with large numbers of agents, they still rely on global mean field information. Even though GMF-Q avoids complete dependence on selecting the optimal action by introducing action distributions, it remains heavily influenced by global information and lacks the ability to finely model the local environment.

## Attention-based mean field multi-agent reinforcement learning

To address the limitations of traditional mean field methods—namely the over-reliance on global averages and the lack of local interaction modeling—recent studies have introduced attention mechanisms into mean field frameworks. Graph Attention Mean Field (GAT-MF) (*Hao et al., 2023*) integrates Graph attention networks with MADDPG, transforming the unweighted mean field into a weighted one that captures varying interaction strengths among agents. This enables more precise modeling in complex environments by dynamically assigning importance to different neighbors.

Similarly, Adaptive Mean Field Q-Learning (AMF-Q) (*Wang et al., 2024b*) incorporates attention into Q-learning to construct an adaptive mean field, allowing agents to flexibly weigh information from nearby agents. By modeling localized influence more accurately,

AMF-Q enhances adaptability and interaction fidelity, especially in scenarios where local neighbor effects dominate agent behavior.

Although these methods can assign different weights to neighboring agents based on the local context, they have two main limitations: First, they lack a global perspective of the environment, relying too heavily on local information and neglecting the influence of the global context. Second, they are constrained by a single-scale modeling approach, which cannot effectively capture the multi-level information in the environment.

To address the above challenges, we propose a multi-scale mean field representation method. This approach introduces both near-field and far-field mean fields to comprehensively capture multi-level interaction information between agents in the environment. Specifically, we obtain information from neighboring agents and apply an attention mechanism to dynamically weigh the neighbor information, forming the near-field information. This process allows for a more accurate reflection of the direct influence neighbors have on the agent, ensuring that the agent focuses on the most relevant neighbors, thereby enhancing the modeling ability of local interactions. At the same time, we construct far-field information using the global action distribution, enabling the agent to gain a richer global perspective. By combining both local and global information, this multi-scale method allows the agent to understand and process interactions at multiple levels, leading to more comprehensive decision making. Through learning multi-scale mean field information, agents can obtain more sufficient and precise environmental information, thereby improving decision accuracy and overall performance.

## PRELIMINARY

This chapter provides a brief introduction to the algorithmic formulas we used.

### Markov decision processes

Due to the problem of path planning in partially observable environments, we adopt the framework of decentralized partially observable Markov Decision Process (Dec-POMDP), whose basic form is expressed as: $\langle S, A_i, P, \Omega_i, O, R, \gamma \rangle$, where $S$ represents the global set of states. $A_i$ denotes the action set of agent $i$, $A = \prod_{i=1}^{M} A_i$ represents the joint action space. $\Omega_i$ is the observation space of agent $i$, and likewise, $\Omega$ denotes the joint observation space. $O : A \times S \rightarrow \Omega$ serves as the observation function of the agents, which can be expressed as the probability $P(o|(a, s))$, where $o \in \Omega$. Given the current state set $S$ and the joint action set $A$, the state transition function and the reward function are defined as follows: $P : S \times A \rightarrow S$, $R : S \times A \rightarrow R$. Here, the state transition function represents the probability $P(s'|(a, s))$ of transitioning from the current state $s \in S$ to the next state $s' \in S$ upon taking action $a \in A$. The long-term cumulative reward starting from time $t_0$ is given by:

$$R_{t_0} = \sum_{t=t_0}^{T} \gamma^{t-t_0} R(s^t, a^t), \tag{1}$$

where $\gamma$ in $[0, 1]$ is the discount factor and $T$ is the maximum length in each episodes.

## Multi-agent Q-learning

MARL, based on RL, assigns a policy to each agent $i$. At each time step, the agent selects an action according to its policy. The future discounted return is used to evaluate the quality of the chosen policy. For a given policy $\pi = X_{i \in \mathcal{N}} \pi_i$ and initial state $s_0 \sim b^0$, the future discounted return for agent $i$ can be expressed as follows:

$$V^i(s; \pi) = \mathbb{E}_{a \sim \pi} \left\{ \sum_{t=0}^{\infty} \gamma^t r^i(s_t, \boldsymbol{a_t}) | s_0 = s \right\}. \tag{2}$$

The $Q$-value function for the multi-agent version can be given by the Bellman equation. During the decision making process, each agent aims to maximize its cumulative reward and will choose the optimal strategy for action. Therefore, for agent ii, the corresponding $Q$-value function $Q^i : S \times \boldsymbol{A} \to \Re$ can be written as follows:

$$Q^i(s, a) = (1 - \alpha)Q^i(s, a) + \alpha \left[ r^i(s, a) + \gamma V^i(s') \right]. \tag{3}$$

In addition to this, the value function of each agent is related to the joint strategy, and the Nash equilibrium is often seen as the goal of learning.

## Nash Q-learning

In MARL, the goal of each agent is to learn an optimal strategy that maximizes its value function. Optimizing the strategy $v^i_\pi$ for agent $i$ requires the joint strategy $\pi$ of all agents. To achieve this, we can borrow the concept of Nash equilibrium from stochastic games to optimize the strategies of the agents. Specifically, a Nash equilibrium is represented by a set of joint strategies $\pi_* \triangleq [\pi_*^1, \ldots, \pi_*^N]$, and for all states $s \in \mathcal{S}, i \in \{1, \ldots, N\}$ and all valid $\pi^i$, the following condition must hold:

$$v^i(s; \pi_*) = v^i(s; \pi_*^i, \pi_*^{-i}) \geq v^i(s; \pi^i, \pi_*^{-i}), \tag{4}$$

where $\pi_*^{-i} \triangleq [\pi_*^1, \ldots, \pi_*^{i-1}, \pi_*^{i+1}, \ldots, \pi_*^N]$ represents the joint strategy of all agents except agent $i$.

As a MARL algorithm, Nash Q-Learning defines a method for computing Nash strategies, consisting of two main steps: 1. Use the Lemke–Howson algorithm to compute the Nash equilibrium for the current stage of the game, and obtain an estimate of the Q-function. 2. Improve the estimate of the Q-function based on the updated Nash equilibrium values, thus optimizing the agent's strategy. Under certain assumptions, the Nash operator $\mathscr{H}^{\text{Nash}}$ can be represented by the following formula:

$$\mathscr{H}^{\text{Nash}} Q(s, a) = \mathbb{E}_{s' \sim p} \left[ r(s, a) + \gamma v^{\text{Nash}}(s') \right], \tag{5}$$

where $Q \triangleq [Q^1, \ldots, Q^N], r(s, a) \triangleq [r^1(s, a), \ldots, r^N(s, a)]$. Eventually, the $Q$-function will converge to the value corresponding to the Nash equilibrium, forming the so-called Nash $Q$-value.

## Mean field approximation

The mean field approach refers to approximating the interactions between many agents as the interaction between an agent and a virtual agent that integrates other agents. If we

assume that the agents in the environment only interact weakly with other agents through the mean field, then the mean field $\mathscr{L}^i \in \delta a^{|\mathcal{O}||\mathcal{A}|}$ can be written as:

$$\mathscr{L}^i = \left(\mu^{-i}(\cdot), \alpha^{-i}(\cdot)\right) = \lim_{N \to \infty} \left( \frac{\sum_{j \neq i} \mathbf{1}(o^j = \cdot)}{N-1}, \frac{\sum_{j \neq i} \mathbf{1}(a^j = \cdot)}{N-1} \right), \tag{6}$$

where $o_j$ and $a_j$ represent the local state and action taken by agent $j$, and $-i$ denotes the set of all agents except for agent $i$.

## METHODOLOGY

This chapter begins with a brief overview of the algorithmic framework we propose. It then presents the formulas used in our method, module by module, and derives the multi-scale mean-field representation formulas. Finally, we provide the pseudocode for the multi-scale mean-field representation method to facilitate a better understanding of its implementation process.

### Framework

For the proposed MSMF-Q method, the encoder consists of two convolutional layers followed by two separate fully connected (FC) layers: one for processing observation maps and one for agent-specific feature vectors. The decoder is composed of two FC layers. And the activation functions are ReLu. All Q-networks used in MSMF-Q and baseline methods follow this architecture unless otherwise specified.

In order to efficiently capture the interaction information of agents at multiple levels, we implemented the framework for training the multi-scale mean field representation method, as shown in Fig. 1. For a given initial environment, we first input each agent's views and features into an encoder for embedding representation. Next, using the blue agent as an example, we extract the action vectors of all agents in the environment at the current time step and average them. This step is consistent with the traditional mean field approach, used to extract global macro-level information from the environment, *i.e.*, far-field information. Then, based on the parameter $k$ (where $k$ is the number of neighbors selected to limit the range of the near field, with $k = 3$ as shown in the figure), we choose the neighbors of the agent. We use distance as a reference, since, generally, the closer an agent is, the stronger its direct influence on the agent. We select the $k$ nearest neighbors of the blue agent to extract its near-field information. The near-field information is dynamically weighted using the attention mechanism, assigning different weights to different agents. Since far-field information has only one value, it is directly passed through the attention mechanism without modification. Finally, the two vectors (near-field and far-field) are concatenated and output as the result of the attention module. This output is concatenated with the embedding representation from the first part and processed through a decoder. The final $Q$-value is then output as a reference for action selection. Finally, the $Q$-value is used to guide the agent in updating its state.

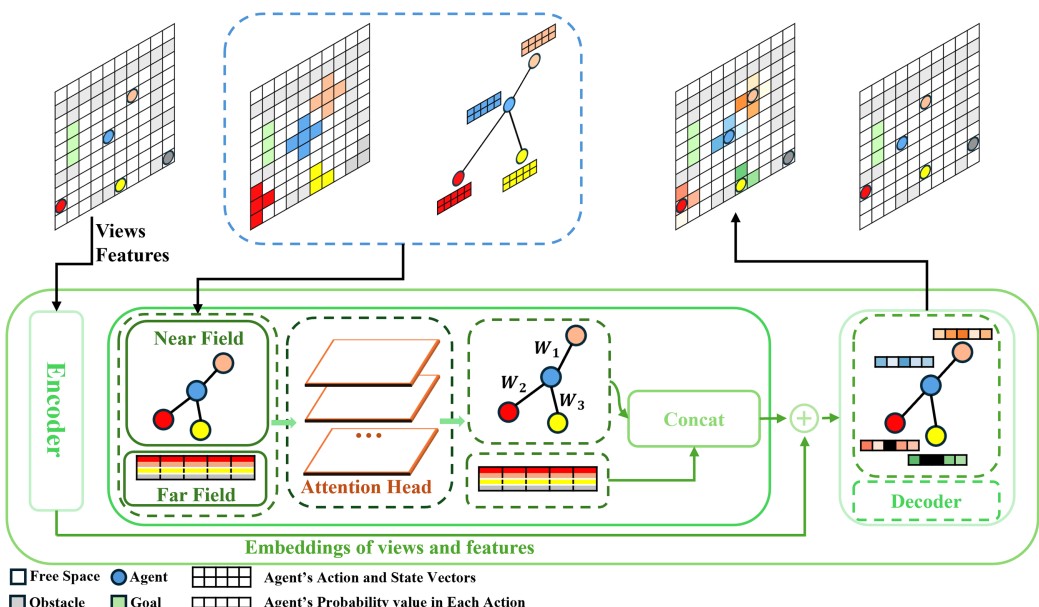

**Figure 1 The image shows the specific implementation framework of the multi-scale mean field representation method.** The top half is an example diagram of agents updating from the current state to the next state, and the bottom half is a diagram of the model framework. The model framework is mainly divided into three parts. The first part is an encoder that processes the global agent views and features. The second part is the attention module that handles the near-field and far-field mean fields at two scales. The third part is a decoder that concatenates the outputs of the first and second parts, performs the calculations and outputs the $Q$-values.

## Multi-scale information extraction and mean field representation

In our work, we adopt a two-scale mean field representation, where the far-field represents the action distribution of all other agents. This can be expressed by the following formula:

$$m_{\text{far}} = \frac{1}{N} \sum_{i=1}^{N} \pi_i(a_i|s_i), \tag{7}$$

where $N$ represents the number of agents in the environment, and $\pi_i(a_i|s_i)$ denotes the probability distribution of agent $i$ selecting action $a_i$ given state $s_i$.

The near-field represents the weighted sum of the action distributions of agent $i's k$ neighbors, with weights computed based on the interaction strength between agents using an attention mechanism. The formula is as follows:

$$m_{\text{near}}^{(i)} = \sum_{i \in \mathcal{N}_k(i)} w_{ij} \cdot \pi_i(a_i|s_i). \tag{8}$$

Finally, the two-scale mean field distributions are fused to obtain the final multi-scale mean field representation:

$$m_{\text{multi}}^{(i)} = m_{\text{far}} + m_{\text{near}}^{(i)}. \tag{9}$$

## Multi-head attention module

We use the attention mechanism to weight the action distributions of neighboring agents and leverage far-field information to learn global context, compensating for the limitations of local neighborhood information.

For the $i$-th query vector $\mathbf{q}_i \in \mathbb{R}^{d_h}$ and the $j$-th key vector $\mathbf{k}_{\text{near},j} \in \mathbb{R}^{d_h}$ of the near field, the attention weight for the $k$ neighbors in the neighborhood is given by:

$$w_{ij}^{\text{near}} = \frac{\exp\left(\frac{\mathbf{q}_i \cdot \mathbf{k}_{\text{near},j}^{\top}}{\sqrt{d_h}}\right)}{\sum_{l=1}^{k} \exp\left(\frac{\mathbf{q}_i \cdot \mathbf{k}_{\text{near},l}^{\top}}{\sqrt{d_h}}\right)}. \tag{10}$$

The final output of the near-field is then obtained as:

$$\mathbf{y}_{\text{near},i} = \sum_{j=1}^{k} w_{ij}^{\text{near}} \cdot \mathbf{v}_{\text{near},j}, \tag{11}$$

where $\mathbf{v}_{\text{near},j} \in \mathbb{R}^{d_h}$ is the value vector for the $j$-th neighbor.

For the far-field, since there is only one global value, the attention weight for the far-field is computed based on the $i$-th query vector $\mathbf{q}_i \in \mathbb{R}^{d_h}$ and the global key vector $\mathbf{k}_{\text{far}} \in \mathbb{R}^{d_h}$, as follows:

$$w_i^{\text{far}} = \frac{\exp\left(\frac{\mathbf{q}_i \cdot \mathbf{k}_{\text{far}}^{\top}}{\sqrt{d_h}}\right)}{\exp\left(\frac{\mathbf{q}_i \cdot \mathbf{k}_{\text{far}}^{\top}}{\sqrt{d_h}}\right)} = 1. \tag{12}$$

The output of the far-field is denoted as:

$$\mathbf{y}_{\text{far},i} = w_i^{\text{far}} \cdot \mathbf{v}_{\text{far}}, \tag{13}$$

where $\mathbf{v}_{\text{far}} \in \mathbb{R}^{d_h}$ is the global value vector.

Finally, by fusing the mean field representations from both scales, we obtain the output vector of the attention module as:

$$\mathbf{y}_i = \mathbf{y}_{\text{near},i} + \mathbf{y}_{\text{far},i}. \tag{14}$$

The information output $y_i$ by the multi-head attention module is processed through a decoder, which generates the model's output, namely the $Q$-value. This $Q$-value reflects the action probability of the agent and will guide the agent in determining its next action.

The algorithm flow is shown in Algorithm 1.

## Mathematical proof

**1. Proof of boundedness for multi-scale mean field error**. *Yang et al.*'s *(2018)* error bound relies on the Lipschitz continuity of single-scale mean fields. Our multi-scale field decomposes into the global field $m_{\text{far}}$ and local field $m_{\text{near}}^{(i)}$ according to Eq. (9).

**Algorithm 1** The basic arithmetic flow of the MSMF-Q algorithm.

**Data:** Environment of MARL *env*
**Result:** The trained models

1  **for** $e = 1 \rightarrow E$ **do**
2      reset env;
3      **while** *step < maxstep* **do**
4          $S_t \leftarrow env$;
5          $V_t, F_t \leftarrow S_t$;
6          For each agent i;
7          Obtain information about the *k* nearest neighbors as initial near-field information;
8          $NF_{(i,t)} \leftarrow topK(S_t)$;
9          Calculate far-field information;
10          $FF_{(i,t)} \leftarrow \frac{1}{N}\sum_{i=1}^{N} \pi_i(a_i|s_i)$;
11          $m_{\text{multi}}^{(i)} \leftarrow Attention(NF_{(i,t)}, FF_{(i,t)})$;
12          $E_v \leftarrow V_t$;
13          $E_f \leftarrow F_t$;
14          $QValues \leftarrow Decoder\left(m_{\text{multi}}^{(i)}, E_v, E_f\right)$;
15          $A_t \leftarrow max(QValues)$;
16          $S_{t+1}, R_t \leftarrow env.step(actions)$;
17          $replayBuffer.store(S_t, S_{t+1}, A_t, R_t, NF_t, FF_t)$;
18      **end**
19      **if** replay buffer size > batch size **then**
20          $q1 \leftarrow Q{-}Network(S_t, NF_t, FF_t)$;
21          $q2 \leftarrow Q{-}Network(S_{t+1}, NF_t, FF_t)$;
22          Obtain the $\hat{q}$ according to Q2 and equation 3;
23          $loss = \frac{\sum (q1-\hat{q})^2 \cdot \text{mask}}{\sum \text{mask}}$;
24          Update the network parameters;
25      **end**
26  **end**

**Error boundedness of global field**. Eq. (7) is identical to *Yang et al.*'s *(2018)* classical mean field. According to *Yang et al.*'s *(2018)* proof, its approximation error satisfies:

$$\|Q_j(s, a_j, m_{\text{far}}) - Q_j(s, \mathbf{a})\| \leq \varepsilon_{\text{far}}, \tag{15}$$

where $\varepsilon_{\text{far}}$ is the Lipschitz constant related to the sparsity of inter-agent interaction intensity.

**Error boundedness of local field**. According to Eqs. (8) and (10), the local field is obtained by weighting neighbors through attention weights $w_{ij}$. Since the attention

weights satisfy normalization and Lipschitz continuity, the approximation error of $m_{\text{near}}^{(i)}$ satisfies:

$$\left\|Q_i\left(s, a_i, m_{\text{near}}^{(i)}\right) - Q_i(s, \mathbf{a})\right\| \leq \varepsilon_{\text{near}}, \tag{16}$$

where $\varepsilon_{\text{near}}$ depends on the neighbor count $k$ and the Lipschitz constant of the attention mechanism.

**Composite error bound for multi-scale field**. According to Eq. (9), since $m_{\text{multi}}^{(i)}$ is a linear combination, its total error satisfies the triangle inequality:

$$\|Q_i(s, a_i, m_{\text{multi}}^{(i)}) - Q_i(s, \mathbf{a})\| \leq \varepsilon_{\text{far}} + \varepsilon_{\text{near}}. \tag{17}$$

Thus, the approximation error of the multi-scale field remains bounded by a constant $\varepsilon = \varepsilon_{\text{far}} + \varepsilon_{\text{near}}$, and $\varepsilon_{\text{near}}$ can be controlled by adjusting neighbor count $k$ (validated in the hyperparameter analysis results of battle game task).

**2. Argument for convergence to nash equilibrium**. According to Eqs. (4) and (5), MSMF-Q's update rules are based on Nash Q-learning. *Yang et al. (2018)* have proven that the operator in Eq. (5) converges to the Nash equilibrium $Q$-value. In MSMF-Q, the $Q$-function update can be expressed as:

$$Q^i(s, a) \leftarrow (1 - \alpha)Q^i(s, a) + \alpha\left[r^i + \gamma \max_{a_i} Q^i\left(s', a_i, m_{\text{multi}}^{(i)}\right)\right]. \tag{18}$$

Since $m_{\text{multi}}^{(i)}$'s error is bounded ($\varepsilon$) and the Nash operator is robust to approximation errors, MSMF-Q's updates converge to a neighborhood of the Nash equilibrium:

$$\lim_{t \to \infty} Q_t = Q_{\text{Nash}}^* + \mathcal{O}(\varepsilon), \tag{19}$$

where $\varepsilon$ is a negligibly small constant (The reward curves verify policy convergence to a high-performance equilibrium).

# EXPERIMENTS

In this chapter, we compare the method we proposed with other methods through experiments conducted in two environments. The experimental setup consists of a server equipped with an Intel(R) Xeon(R) Gold 6226R CPU @ 2.90 GHz and Nvidia 3090 GPUs. The experimental results show that our method performs better to some extent.

## Settings

The following hyperparameters were used throughout all experiments: a learning rate of 0.001 with the Adam optimizer, batch size of 64, and a target Q-network update frequency of every 10 steps. The discount factor $\gamma$ was set to 0.95. We adopted an $\varepsilon$-greedy exploration strategy, with $\varepsilon$ linearly annealed from 1.0 to 0.1 over the course of training. A replay buffer was used to store transitions, with a capacity of $2^{10}$ tuples.

For computing near-field interactions, we used the Euclidean distance between agents' current coordinates to select the $k$ nearest neighbors. The real-world grid task is based on

**Table 1  The rewards of battle game task.**

| Actions | Reward |
|---|---|
| Moving | −0.005 |
| Attacking an enemy | 0.200 |
| Killing an enemy | 5.000 |
| Attacking an empty grid | −0.100 |
| Being attacked or killed | −0.100 |

**Table 2  The rewards of real grid world task.**

| Actions | Reward |
|---|---|
| Moving or waiting (up, down, left, right, or staying) | −0.3 |
| Collisions incurred | −2.0 |
| Reaching an exit | 5.0 |

internal layout data from railway station. The original station floor plan was abstracted into a $120 \times 80$ 2D grid map, where each cell represents a navigable space or an obstacle.

**Scenario 1: Battle game task.** The first experiment was conducted in the MAgent system under the Pettingzoo package, a widely used environment for MARL. In this scenario, two teams of agents, each employing different MARL methods, compete against each other in a grid world. The objective of the game is to eliminate the opposing team, with agents capable of moving in four directions (up, down, left, or right) or attacking adjacent grids. The game dynamics are designed to simulate competitive decision making and strategy optimization, which are common challenges in distributed decision making systems such as resource allocation and competitive market simulations. In this experiment, we set the number of agents to 128 *vs.* 128, with rewards defined for various actions: moving (−0.005), attacking an enemy (0.200), killing an enemy (5.000), attacking an empty grid (−0.100), and being attacked or killed (−0.100). Specific values are shown in Table 1. These settings mimic real-time decision making in high-stakes environments where agents must learn to optimize both offensive and defensive strategies in the presence of competitors. In addition, in this scenario, we set the number of neighbors that need to be calculated for attention to be $k = 16$, and the reason is given by the hyperparameter analysis experiment.

**Scenario 2: Real grid world task.** The second experiment applied the MSMF-Q method to a robotic path planning scenario, simulating a crowd evacuation in a real-world environment. Using station data from railway station, we constructed a two-dimensional grid model of the station and mapped it to a 2D grid world for agent navigation. This experiment is highly relevant to robotic scheduling and smart city management, where path planning plays a critical role in optimizing the movement of robots or automated crowd control in large-scale spaces. In this simulation, 512 agents were randomly assigned positions on the grid, and their objective was to reach one of the exits to successfully evacuate the station. The map size for each layer was set to $120 \times 80$, representing the

station layout. Reward values were set as follows: moving or waiting (up, down, left, right, or staying) ($-0.3$), collisions incurred ($-2.0$), and successfully reaching an exit ($5.0$). Specific values are shown in Table 2. This scenario evaluates the method's ability to manage large-scale agent coordination and path optimization in real-world robotic applications, focusing on efficient route planning and dynamic decision making under various environmental conditions. Furthermore, hyperparameter configuration aligns with spatial agent density. For the evacuation task, neighborhood size $k$ scales with population size $N$—set to $k = 8(N = 256)$, $k = 12(N = 512)$, and $k = 16(N = 1024)$—based on the density-to-$k$ ratio.

All results presented are averaged over five independent runs using distinct random seeds. The reported standard deviations and success/win rates are computed based on these runs.

## Baseline

We trained multiple mean field methods on the MAgent platform and our constructed real-world grid environment. On the MAgent platform, different MARL algorithms were used to control teams for competitive battles. For the evacuation task, the success rate of agents after 512 time steps was recorded. The algorithms involved are as follows: **MF-Q** (*Yang et al., 2018*): MF-Q was one of the first methods to apply mean field theory to MARL. It abstracts the action distribution of all agents into a virtual agent, transforming interactions between agents into interactions between an agent and the virtual agent. This significantly reduces the computational complexity caused by the large number of agents in MARL. **GMF-Q** (*Guo et al., 2019*): This method combines the Boltzmann strategy with Q-learning. Agents' action selection not only aims to maximize the current $Q$-value but also considers the probability distribution of other actions. This strategy helps avoid local optima and enhances exploration. **AMF-Q** (*Wang et al., 2024b*): Building on traditional mean field methods, AMF-Q introduces an attention mechanism. This allows agents to dynamically consider the influence of neighboring agents on their decisions. The dynamic weighting mechanism effectively handles the impact of heterogeneous environments on agents. **MSMF-Q:** We proposed the method that uses an attention mechanism to dynamically weight neighbors and integrates global mean field information to account for global action distributions. This approach, combining near-field and far-field information, effectively captures multi-level environmental details.

## DISCUSSION

The observed performance advantages of MSMF-Q over all baselines, demonstrated by consistently higher average rewards, win rates, and success rates across multiple independent runs and scenarios, are substantial and statistically significant based on the clear separation of mean performance and standard deviation bounds.

**Description of indicators:** In the battle game, the reward value refers to the total return value of all agents. The convergence speed and stability of the reward curve reflect the efficiency of the algorithm. A faster and more stable convergence indicates better algorithm performance. The win rate refers to the proportion of victories in total matches

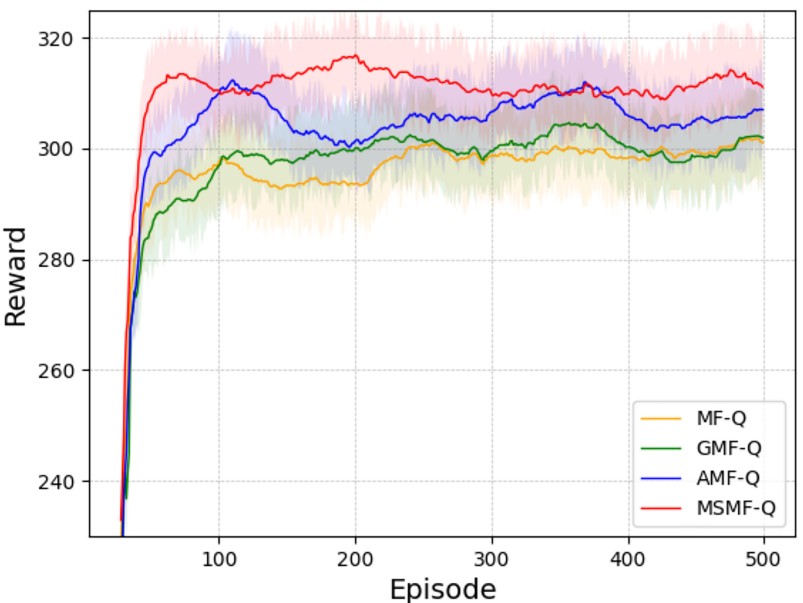

**Figure 2** **The average rewards of battle game task.**

when two teams, each controlled by different algorithms, compete. A higher win rate suggests stronger algorithm performance. In the 2D grid path finding scenario, the agent density distribution refers to the spatial distribution of agents after simulating for a certain period of time. The more evenly agents are distributed across multiple target points, the better the algorithm's performance is typically considered. The success rate measures whether the algorithm can successfully find the target point and guide agents to reach it within a fixed operating time. A higher success rate indicates stronger path planning ability of the algorithm.

**Scenario 1: Battle game task.** After 2,000 rounds of self-play training, we used different models for adversarial simulations. Figure 2 shows the average reward training curves. In the figure presenting learning curves with shaded regions, the shaded region represents the standard deviation ($\sigma$) across multiple independent runs. This shading visually depicts the variability in the performance metric around the mean value (solid line) at each training episode.

From the figure, we can analyze that GMF-Q converges slightly better than MF-Q. This is mainly because GMF-Q, based on MF-Q, incorporates the Boltzmann strategy, which considers the action distribution of agents rather than directly choosing the action with the highest $Q$-value for decision making. AMF-Q, due to the addition of the attention mechanism, enables agents to have a more flexible weighting mechanism for neighboring agents. It does not equally attend to all other agents but rather focuses more on the neighbors that have a significant impact on its own decisions. This improves the learning effect to some extent, leading to better convergence. Our proposed method, by introducing the attention mechanism and multi-scale mean field representation, not only focuses on near-field information that has a significant impact on the agent itself but also learns global

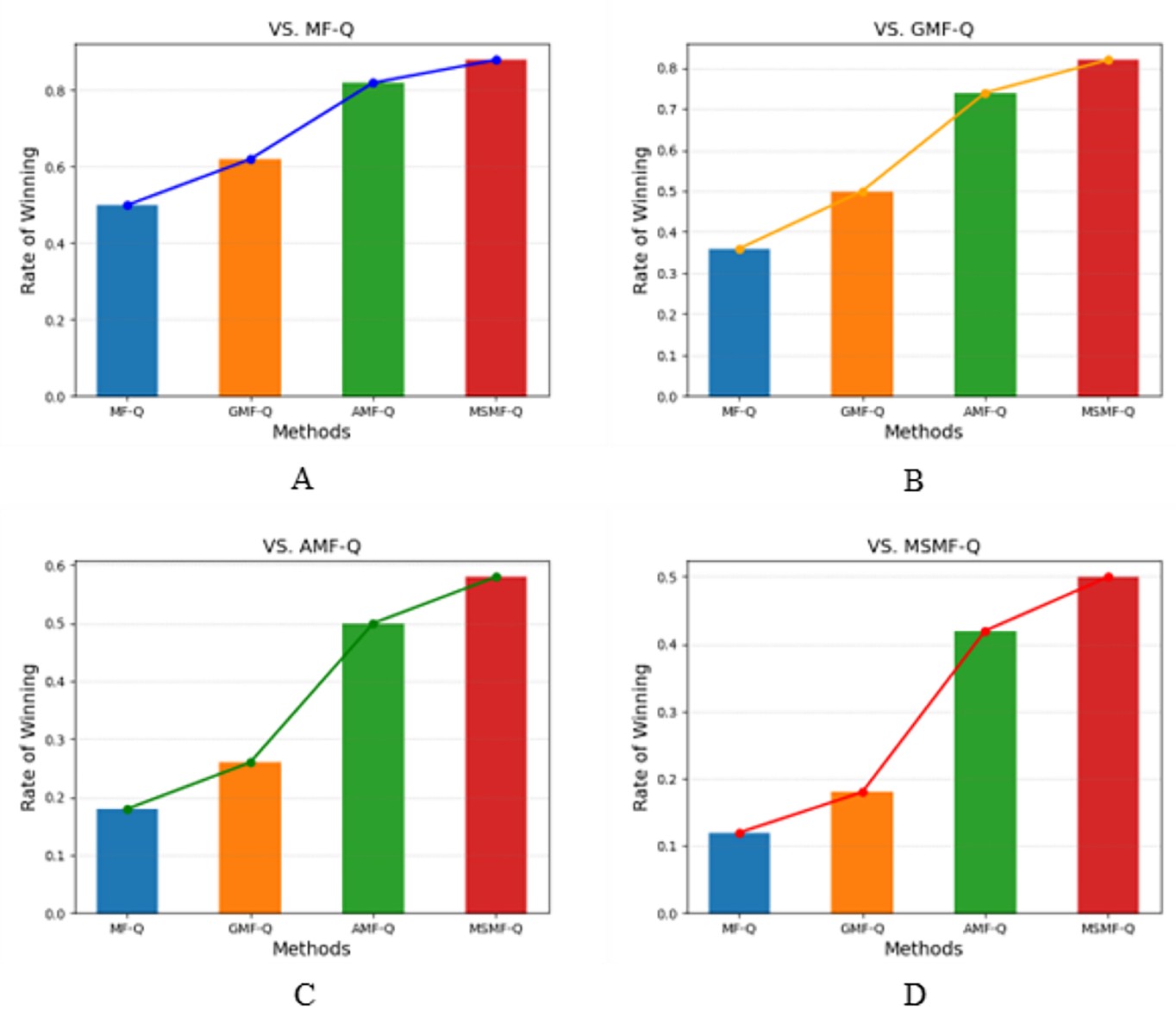

**Figure 3 The success rate of battle game task.** (1) Blue bars represent the success rate of MF-Q *vs.* each of the four algorithms (A–D), and the blue line represents the success rates against MF-Q. (2) Orange bars represent the success rate of GMF-Q *vs.* each of the four algorithms (A–D), and the orange line represents the success rates against GMF-Q. (3) Green bars represent the success rate of AMF-Q *vs.* each of the four algorithms (A–D), and the green line represents the success rates against AMF-Q. (4) Red bars represent the success rate of MSMF-Q *vs.* each of the four algorithms (A–D), and the red line represents the success rates against MSMF-Q.

context information. This allows the agent to better understand global information and achieve better convergence.

Figure 3 and Table 3 presents the results of various algorithms in a competitive game scenario. Specifically, Fig. 3A shows the success rates of teams controlled by four different algorithms after 50 rounds of competition against a team controlled by the MF-Q algorithm. Similarly, Figs. 3B, 3C, and 3D display the results of the four algorithms

**Table 3 The success rate of battle game task.** The bold indicates the best-performing metrics achieved in a comparative analysis of the algorithms.

| Methods | VS. MF-Q | VS. GMF-Q | VS. AMF-Q | VS. MSMF-Q |
|---------|----------|-----------|-----------|------------|
| MF-Q | 0.50 | 0.36 | 0.18 | 0.12 |
| GMF-Q | 0.62 | 0.50 | 0.26 | 0.18 |
| AMF-Q | 0.82 | 0.74 | 0.50 | 0.42 |
| MSMF-Q | **0.88** | **0.82** | **0.58** | **0.50** |

competing against teams controlled by the GMF-Q, AMF-Q, and MSMF-Q algorithms, respectively.

From the figures, it is evident that the MSMF-Q method consistently achieves a high win rate in all competitive settings, outperforming the other three methods (MF-Q, GMF-Q, and AMF-Q) with a significantly higher success rate. This clearly demonstrates the superior performance of our proposed multi-scale mean-field method (MSMF-Q). Specifically, MSMF-Q achieves the highest win rate, indicating its stronger learning capacity and superior training outcomes. The AMF-Q algorithm ranks second in terms of win rate, showing that the incorporation of the attention mechanism allows AMF-Q to better adapt to dynamic changes in local information. In contrast, GMF-Q ranks third; despite enhancing strategy learning through the introduction of action distributions, it still lags behind the other methods in terms of overall performance. Finally, MF-Q achieves the lowest success rate, reflecting the inherent limitations of traditional single-scale mean-field methods when applied to complex competitive tasks. These experimental results further validate the effectiveness of our approach, highlighting that the multi-scale mean-field method can capture the complex interaction dynamics between agents more comprehensively, thereby improving the overall performance of the algorithm.

Additionally, in the experiment, we conducted a hyperparameter analysis to determine the optimal value of $k$, which represents the number of neighboring agents considered in the near-field information. Since the choice of $k$ directly affects the computational efficiency of the attention mechanism, we analyzed how different values of $k$ impact both the average maximum reward return and the algorithm's computational efficiency.

The results are shown in Figs. 4, 5 and Table 4. Figure 4 illustrates the curves showing the impact of different $k$ values on algorithm efficiency and the final average maximum reward. From these curves, we observe that, within a certain range of $k$, a larger value of $k$ leads to a higher final average maximum reward. This is because, as the agent considers more neighboring agents' information, the attention mechanism dynamically assigns weights to more neighbors, which helps the agent learn more detailed interaction information. However, when $k$ exceeds a certain threshold, such as when $k$ exceeds 22 in Fig. 4, the near-field encompasses too many neighbors, leading the attention mechanism to focus on noise, ultimately resulting in less optimal performance. In terms of computational efficiency, we calculated the average training time per episode for different values of $k$. As expected, the larger the value of $k$, the more information the attention mechanism needs to process, which results in a decrease in computational efficiency.

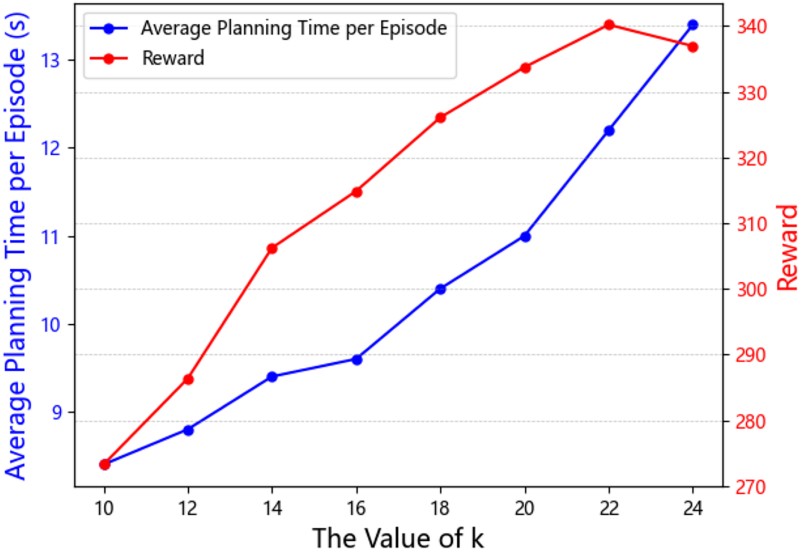

**Figure 4  The hyperparameter analysis results of battle game task.**

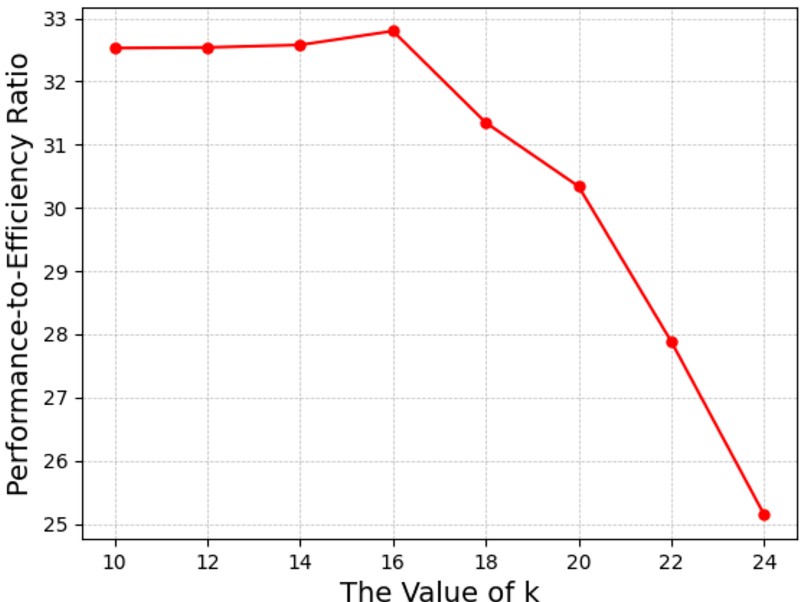

**Figure 5  The performance-to-efficiency ratio of battle game task.**

**Table 4  Hyperparameter analysis.** The bold indicates the highest (optimal) values obtained for the Reward and PER metrics across the range of k-values tested in our benchmarking study.

| k | 10 | 12 | 14 | 16 | 18 | 20 | 22 | 24 |
|---|---|---|---|---|---|---|---|---|
| Time | 8.4 | 8.8 | 9.4 | 9.6 | 10.4 | 11.0 | 12.2 | 13.4 |
| Reward | 273.26 | 286.35 | 306.22 | 314.86 | 326.07 | 333.72 | **340.18** | 336.94 |
| PER | 32.53 | 32.54 | 32.58 | **32.80** | 31.35 | 30.34 | 27.88 | 25.14 |

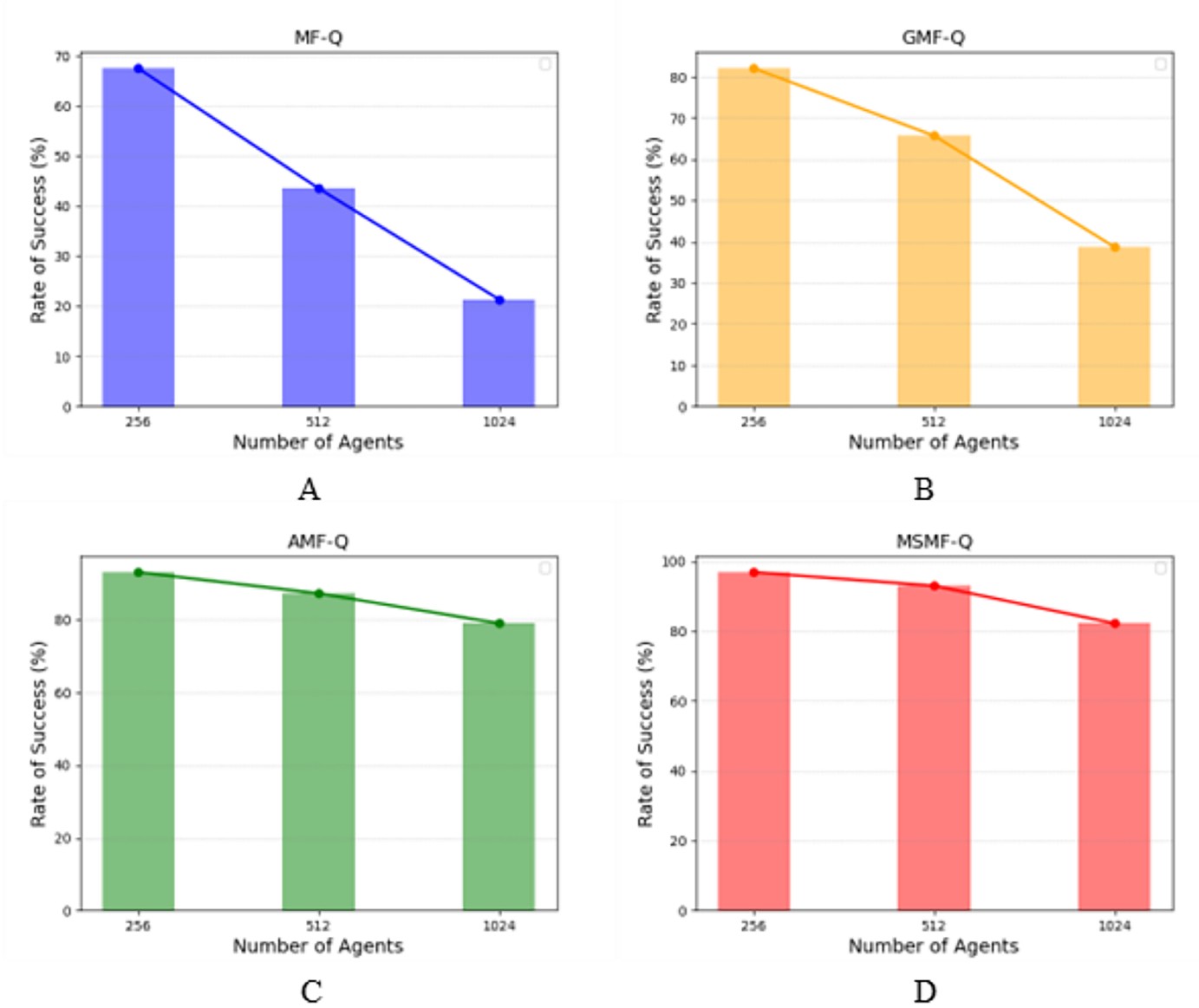

**Figure 6 The success rate of real grid world task.** (A) Blue bars and blue line represent the success rate of MF-Q. (B) Orange bars and orange line represent the success rate of GMF-Q. (C) Green bars and green line represent the success rate of AMF-Q. (D) Red bars and red line represent the success rate of MSMF-Q.

To balance these two factors, we introduced a performance-to-efficiency ratio (PER) as a reference for hyperparameter analysis. This ratio is defined as the average maximum reward divided by the average training time per episode. The higher the ratio, the more suitable the corresponding $k$ value. The results, shown in Fig. 5, indicate that when $k = 16$, the performance-to-efficiency ratio is maximized. Although Fig. 4 shows that $k = 22$ yields the highest average maximum reward, the computational efficiency also declines, which contradicts the initial design goal of the mean field approach. Therefore, in this experiment, we selected $k = 16$ as the optimal value.

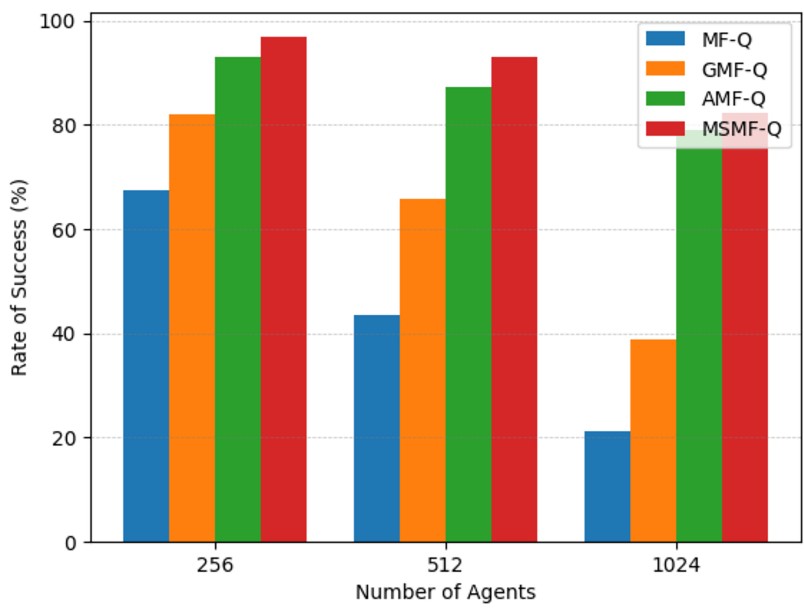

**Figure 7** **The results of real grid world task.**

**Scenario 2: Real grid world task.** Through training, we obtained the evacuation success rates of different algorithms when the number of agents was set to 256, 512, and 1,024. As shown in Fig. 6, we computed the proportion of agents that successfully reached the target point within a fixed number of time steps, which is referred to as the success rate. Specifically, Fig. 6A represents the success rate of agents controlled by the MF-Q algorithm in reaching the exit after a certain number of time steps. Similarly, Figs. 6B, 6C, and 6D show the success rates of the GMF-Q, AMF-Q, and our proposed MSMF-Q algorithms under the same conditions, respectively.

The experimental results across different numbers of agents reveal an important trend: as the number of agents increases, the success rates for both the MF-Q and GMF-Q algorithms decline more rapidly, indicating a lack of scalability for these algorithms with respect to agent count. In contrast, the AMF-Q algorithm and our proposed method, due to their adaptive nature, demonstrate better control capabilities. They maintain higher success rates when agents are more sparsely distributed across the map or when the number of agents fluctuates rapidly, showcasing a better ability to handle such dynamic changes.

In order to facilitate the comparison between the effects of different algorithms, we visualized the experimental results of different algorithms in the same figure. As shown in Fig. 7, from the success rate of each algorithm, it is evident that MF-Q, due to its more singular objective, leads to fewer agents reaching the target in the same time. GMF-Q, with the addition of the Boltzmann strategy, has a higher success rate than MF-Q, but due to its strong dependence on global action distributions, its success rate is still lower than that of AMF-Q, which is an adaptive mean-field approach. AMF-Q, due to its reliance on local information, results in a generally lower success rate compared to our proposed method.

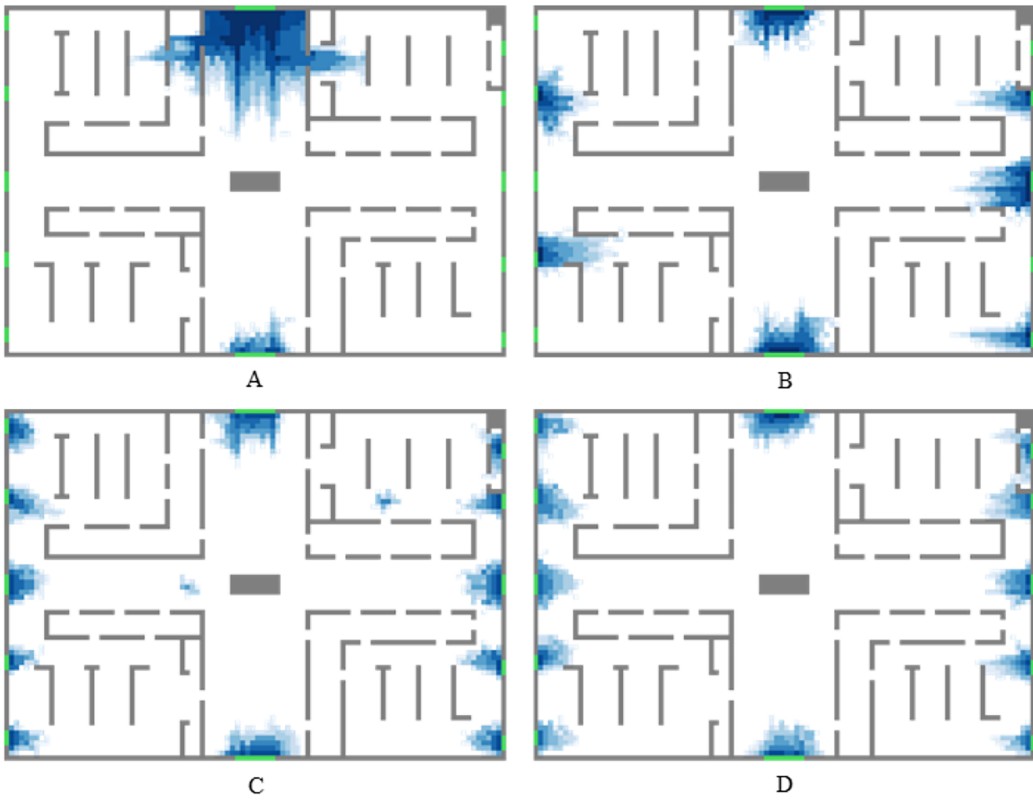

**Figure 8 The result of various algorithms in real grid world task.**

Our proposed multi-scale mean-field method, guided by near-field information, allows agents to quickly find reasonable local paths. Furthermore, under the guidance of far-field information, it avoids local optima, leading to a more balanced distribution across multiple target exits.

In order to verify the observations we made above, we performed density statistics in an environment of 512 intelligences. Figure 8 shows the agent density map of a specific region on the map. Among them, Figs. 8A, 8B, 8C and 8D represent the results of the methods MF-Q, GMF-Q, AMF-Q, and our proposed MSMF-Q, respectively. From the figure, it can be seen that with our proposed method, the distribution of agents around the exits is more uniform, and the areas with higher density are located near the exits. In contrast, the map trained with MF-Q shows a strong tendency for agents to cluster around a single exit. This might be due to the homogeneity assumption in MF-Q, which leads to a strong global trend, making individuals more likely to be guided by the overall system. GMF-Q, by incorporating the Boltzmann strategy, action distribution, and randomized strategy, alleviates this phenomenon to some extent. However, it is still influenced by global actions, resulting in uneven agent distribution. AMF-Q, with the attention mechanism dynamically weighting neighboring agents, still depends on local information for decision making, leading to occasional group clustering. The experimental results show that the causes we have analysed are completely reliable.

**Table 5 Average reward and standard deviation for each method over 5 runs.**

| Method | Mean reward | Std. dev |
|---|---|---|
| MF-Q | 301.71 | 0.43 |
| GMF-Q | 304.70 | 0.38 |
| AMF-Q | 312.49 | 1.04 |
| MSMF-Q | 316.98 | 0.36 |

**Table 6 Statistical significance results from independent t-tests.**

| Comparison | p-value | Significance level |
|---|---|---|
| MSMF-Q *vs* AMF-Q | 0.021 | $p < 0.05$ |
| MSMF-Q *vs* GMF-Q | 0.0006 | $p < 0.01$ |
| MSMF-Q *vs* MF-Q | < 0.0001 | $p < 0.001$ |

**Statistical significance analysis.** To validate the robustness of performance differences between MSMF-Q and other methods, we conducted independent two-sample t-tests based on five repeated experimental runs with slight stochastic variation. The results are summarized in Tables 5 and 6. Notably, AMF-Q exhibited relatively large performance variance due to its reliance on local neighbors. Despite this, MSMF-Q consistently achieved higher average rewards with statistical significance. Specifically, MSMF-Q significantly outperformed AMF-Q ($p = 0.021$), GMF-Q ($p = 0.0006$), and MF-Q ($p = 0.0001$), confirming that the observed advantages are not due to randomness, but reflect consistent improvements.

## CONCLUSION

In this study, we propose a multi-scale mean field representation method for MARL to overcome the limitations of traditional mean field approaches. Conventional mean field methods typically focus on agent interactions at a single scale, often overlooking the intricate multi-level interactions between agents and the environment. To address this gap, our approach integrates multi-level agent-environment interaction information, enhancing the performance of reinforcement learning in large-scale, multi-agent systems. Specifically, the proposed method combines both near-field and far-field mean field representations. The near-field scale emphasizes dynamic interactions with local neighbors, where more influential neighbors are assigned higher weights, ensuring the agent attends to the most relevant interactions. In contrast, the far-field scale captures global action distributions, modeling the macro-level effects and overall behavior of the system. By learning these multi-scale mean fields, we enable more accurate simulation of agent decision making under complex, multi-level interactions. Extensive experimental validation across various scenarios demonstrates the superior effectiveness and adaptability of our method. Compared to traditional mean field methods, our approach is better equipped to handle complex, heterogeneous environments, offering improved local optimization and strategic global guidance. This work not only enhances agent behavior at

the local level but also provides theoretical and practical support for advancing MARL in large-scale, real-world applications.

While the current study establishes adaptive neighborhood scaling *via* density-k mapping, future research will explore theoretically grounded scaling laws for attention-based multi-agent systems. Future work will formalize theoretically grounded scaling laws while extending to ultra-scale heterogeneous systems. This paves the way for certifiable multi-agent coordination in exponentially growing state spaces.

## ACKNOWLEDGEMENTS

We sincerely thank all those who provided support and assistance during the research process for the completion of this work.

### Funding

This work was supported by the "Pioneer" and "Leading Goose" R&D Program of Zhejiang under Grant 2024C01214, by the National Natural Science Foundation of China under Grant 62476247, 62072409 and 62073295, and by the Zhejiang Provincial Natural Science Foundation under Grant LR21F020003. There was no additional external funding received for this study. The funders had no role in study design, data collection and analysis, decision to publish, or preparation of the manuscript.

### Grant Disclosures

The following grant information was disclosed by the authors:
"Pioneer" and "Leading Goose" R&D Program of Zhejiang: 2024C01214.
National Natural Science Foundation of China: 62476247, 62072409 and 62073295.
Zhejiang Provincial Natural Science Foundation: LR21F020003.

### Competing Interests

Xiangjie Kong is an Academic Editor for PeerJ.

### Author Contributions

- Guowen Li conceived and designed the experiments, performed the experiments, analyzed the data, performed the computation work, prepared figures and/or tables, authored or reviewed drafts of the article, and approved the final draft.
- Jiaxin Du conceived and designed the experiments, analyzed the data, authored or reviewed drafts of the article, and approved the final draft.
- Zhenzhen Zhao conceived and designed the experiments, performed the experiments, analyzed the data, prepared figures and/or tables, authored or reviewed drafts of the article, and approved the final draft.
- Guojiang Shen conceived and designed the experiments, authored or reviewed drafts of the article, and approved the final draft.
- Xiangjie Kong conceived and designed the experiments, authored or reviewed drafts of the article, and approved the final draft.

## Data Availability

The data is available in the Supplemental File.

Additional experimental environment data is available at GitHub:
https://github.com/Farama-Foundation/MAgent2

## Supplemental Information

Supplemental information for this article can be found online at http://dx.doi.org/10.7717/peerj-cs.3222#supplemental-information.

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
