# Peer review of "Multi-scale mean field learning for adaptive decision making in multi-agent systems"

_PeerJ Computer Science, doi:10.7717/peerj-cs.3222_

## Round 0.1 · original submission · Major Revisions

Please consider the comments provided by reviewers #1 and #2, which will improve the clarity of the article. Reviewer #2 outlined several improvements that will facilitate the readability of the article. Most importantly, it is imperative to address the comments by reviewer #3, specifically pertaining to the following points:
1) The lack of theoretical justification for the mean field approximation
2) Important prior literature that is neither cited nor compared to
4) Incomplete Experimental results
5) Substantiate claims regarding heterogeneous agent environments

The revised submission must include a detailed summary of the changes made in response to the reviewers' comments.

Reviewer 1 ·

Basic reporting

The manuscript is overall clearly written, structurally sound, and well-organized into logical sections. The literature review is extensive and properly contextualizes the proposed method within current research trends in multi-agent reinforcement learning (MARL), particularly mean-field methods.

However, there are some points for improvement:

- Lines 39-44: The sentence describing the challenges in MARL due to "explosive growth" could be made clearer by explicitly stating what specific dimensions of interactions are problematic.

- Lines 88-165 are extensive but somewhat repetitive. Reducing redundancy and clearly highlighting the gap that your method fills would significantly strengthen this section.

Experimental design

The experimental setup is solid, with clearly defined scenarios: a competitive battle simulation and a real-world grid evacuation scenario. These two scenarios effectively test both the general applicability and specific usefulness of the proposed method.

Points of improvement are:

- Provide more specific explanations for the reward settings used for each particular case, particularly in the "Real grid world task". For instance, the penalty values of -0.3 for movement and -2.0 for bumping are not explained. How were these values determined? A citation of a preceding study or sensitivity analysis would be more believable.

- There has been a good comparison with contemporary methods (MF-Q, GMF-Q, AMF-Q). However, the reason for selecting these particular methods must be reiterated briefly in a concise manner here, highlighting precisely why these methods are fair benchmarks.

Validity of the findings

The results presented are coherent and valid. The article well illustrates the advantage of including multi-scale interactions in mean field methods. The inclusion of the attention mechanism clearly enhances the model's dynamic capacity to adapt to evolving environments.

Further testing is recommended:

- Include a discussion of the limitations of your experimental findings, especially your method's relevance to other complicated real-world scenarios beyond your two test cases.

- Although you did hyperparameter tuning, the method of selecting the most performance-efficient hyperparameter (k=16) justifies itself primarily on an efficiency-performance trade-off basis. It would be better to have more than this based on other statistical cross-checking or robustness checks (e.g., sensitivity analysis).

Additional comments

In terms of technical quality, figures that appear within the manuscript are mostly legible, readable, and fitting to the manuscript's arguments and findings. However, some issues are also present, which require attention.

In fact, almost all the figures appear slightly pixelated and blurry when zoomed in. To ensure readability, the authors should provide a high-resolution vector graphic.

Reviewer 2 ·

Basic reporting

No Comment

Experimental design

No Comment

Validity of the findings

No Comment

Additional comments

Major Comments & Areas for Improvement:
1. Statistical Rigor: The manuscript should enhance its statistical reporting to fully substantiate the claims of the proposed method's efficacy. While graphical representations (e.g., Figure 2 and others) may include shaded regions indicating variance, the following details are crucial for a robust interpretation of the results:
a. Explicit Definition of Variance Metrics: The figure captions or main text must explicitly state what any shaded regions or error bars represent (e.g., standard deviation, standard error, confidence interval).
b. Number of Runs/Seeds: The exact number of independent experimental runs or different random seeds used to generate averages and variance measures must be clearly stated, typically in the methodology or experimental setup section. This information is fundamental for assessing statistical robustness.
Recommendation: Ensure all reported performance metrics are accompanied by the number of independent runs/seeds. Explicitly define all variance indicators used in the figures. To strengthen claims of superiority, consider applying and reporting appropriate statistical tests to formally compare the performance of different algorithms.

2. Replicability and Methodological Clarity: For the scientific community to verify and build upon this work, comprehensive details regarding the experimental setup and model architectures are essential. The current manuscript lacks some specifics that would be necessary for full replication.
Recommendations: Provide more exhaustive details, potentially in an Appendix or a dedicated subsection within the main text:
a. Specify the neural network architectures (number of layers, units per layer, activation functions) for the encoder, decoder, and Q-networks used in MSMF-Q and all baseline methods.
b. List all relevant hyperparameters used during training. This includes, but is not limited to, learning rates, optimizer details, batch sizes, replay buffer capacities, exploration strategies and their parameters (e.g., epsilon in epsilon-greedy), and the specific discount factor (γ) employed.
c. Clearly define the distance metric used for selecting the k-nearest neighbors in the near-field computation.
d. Regarding the real grid world task (line 283 mentions data from Hangzhou West Railway Station), provide more details about this dataset. Discuss its characteristics, how it was processed or abstracted for the simulation, and its source or availability, if applicable.

3. Clarification and Justification of Hyperparameter Choices in Main Experiments: The paper includes a hyperparameter analysis for k (number of neighbors), concluding that k=16 is optimal for the battle game task based on the PER (lines 370-373, Table 4). However, it is not explicitly clear if this optimized value of k=16 was consistently used for all reported results in the battle game scenario, or what value of k was used for the real grid world task and the baseline comparisons. Figure 1 uses k=3 as an illustrative example, which could cause confusion if not properly contextualized.
Recommendation: Provide some justification for the choice of hyperparameters like k.

4. Novelty Articulation: While the constituent components of the proposed method (mean-field approximations, attention mechanisms) are established, the core contribution lies in the specific multi-scale formulation that combines near-field attention with a far-field global average. The manuscript could more sharply delineate the novelty and advantages of this particular combination, especially in contrast to alternative approaches for integrating local and global information, such as hierarchical attention mechanisms or other multi-scale representations.
Recommendation: Enhance the discussion to crisply articulate why this specific architectural choice is particularly effective and novel for the target problem. A more direct comparison with conceptual alternatives could strengthen this aspect.

5. Computational Cost Analysis: A practical assessment of any new MARL algorithm should consider its computational demands. The paper discusses the performance-to-efficiency ratio (PER) concerning the hyperparameter k but lacks a broader comparison of the computational overhead of MSMF-Q against the baseline methods, particularly AMF-Q, which also employs attention mechanisms.
Recommendation: Include a brief discussion or analysis of the computational costs (e.g., training time per episode, number of parameters, FLOPs if feasible) of MSMF-Q relative to the baselines. This will provide a more complete picture of its practical trade-offs.

6. Discussion of Limitations and Future Work: A comprehensive research paper typically acknowledges the limitations of the proposed method and outlines potential avenues for future research. This element is currently missing from the conclusion.
Recommendation: Add a short subsection to the Conclusion discussing any identified limitations of the MSMF-Q approach (e.g., sensitivity to certain hyperparameters, assumptions made, scalability limits not yet tested) and suggest promising directions for future work.

7. Language Polish and Minor Clarifications: While the manuscript is generally understandable, several instances of phrasing could be refined for improved clarity, precision, and adherence to academic writing conventions.
Recommendations: Conduct a thorough proofread, ideally with the assistance of a professional editing service or a native English speaker proficient in the field. Specific examples include:
a. Line 31-32: "significantly driving the adoption of intelligent technologies" could be phrased more directly or concisely.
b. Line 62: The term "operability" in "ensuring both its operability and theoretical grounding" is somewhat unconventional in this context. Consider alternatives like "practical applicability," "implementability," or "empirical validation."
c. Line 62: The abbreviation "MSMF-Q" is introduced without its full form being explicitly stated beforehand in the main text. Ensure all acronyms are defined upon their first use.

Reviewer 3 ·

Basic reporting

The paper's key insight is to include multi-level interaction-based mean field modelling, where agents nearby and agents further away are modelled differently at multiple different scales (called near-field and far-field mean fields).

I have the following concerns about the paper:

1) No theoretical justification for the mean field approximation:

One major problem I have with the paper is that Equation 9 is proposing a linear summation of two different mean field formulations, but there is no theoretical guarantee that the approximation error induced by the two-scale mean fields will be bounded. For example, Yang et al. [1] prove that their formulation of the mean field approximation guarantees that the approximation error between the multi-agent Q function and the mean field Q function is bounded by a small Lipschitz constant, which can be potentially neglected. Further, they prove that Q updates using their mean field Q function converge to the Nash equilibrium. I would expect to see similar theoretical guarantees in this work, since it is also introducing a mean field approximation similar to Yang et al. Lack of these theoretical guarantees makes me wonder if the approximation error is potentially unbounded. The mean field approximation introduced in this work is not shown to be principled, which is a major problem.

2) Important prior literature not cited and compared:

This paper misses an important reference [2]. The work in [2] uses a very similar approach to this paper, where [2] introduces a hierarchical framework for mean field learning, weighting the impact of other agents on multiple different scales, exactly the same as the motivation in this paper. The authors should have cited [2] and compared their work to the algorithm in [2]. This is another major limitation of this paper. Given [2], I do not believe that this work has sufficient novelty.

[3] Important baselines missing:

Though this work cites [3], it should have been used as a baseline in the experiments. Gat-MF introduced by [3] uses an attention mechanism and mean field learning, the same as the motivation in this work.

[4] Experimental results are incomplete:

Several results in the experiments are close, and a statistical significance test should have been performed. For example, in Figure 2, the performance of MSMF-Q is very close to the performance of AMF-Q, and it is not clear if the comparisons are indeed statistically significant. Similarly, several performances are close in Figure 3, and a statistical significance test is required.

[5] Claims of heterogeneous agent environments:

The paper mentions that it extends mean field learning to environments requiring heterogeneous agent decision-making multiple times (abstract, introduction, conclusion, etc.), however, I do not see any evidence for this in the experiments. All the experimental settings used in the paper have fully homogeneous agents, and I am not sure how MSMF-Q applies to heterogeneous agent settings.

[1] Yang, Yaodong, et al. "Mean field multi-agent reinforcement learning." International conference on machine learning. PMLR, 2018.

[2] Yu, Chao. "Hierarchical mean-field deep reinforcement learning for large-scale multiagent systems." Proceedings of the AAAI Conference on Artificial Intelligence. Vol. 37. No. 10. 2023.

[3] Hao, Qianyue, et al. "Gat-mf: Graph attention mean field for very large scale multi-agent reinforcement learning." Proceedings of the 29th ACM SIGKDD Conference on Knowledge Discovery and Data Mining. 2023.

Experimental design

The experimental design is sound, but important baselines are missing. Also, certain central claims of the paper are not supported by the experiments, as I have mentioned in the previous section.

Since the paper does not report the values for all hyperparameters used in the algorithms and does not open-source the code, the central experiments in the paper are not reproducible in the current form.

Validity of the findings

There are important concerns regarding novelty, statistical significance, and reproducibility, as stated in my response to the previous two sections.

---

## Round 0.2 · accepted · Accept

The authors have addressed reviewers' concerns and have made necessary modifications. The manuscript in its current form is ready for publication.

Reviewer 1 ·

Basic reporting

The manuscript is written in clear, professional English and maintains an unambiguous style throughout. The introduction and related work sections provide sufficient background and context with appropriate references to the state of the art. The structure of the article is coherent and follows professional standards, with figures and tables that are well prepared and appropriately referenced. The results and supporting data are clearly presented, making the study self-contained and relevant to the stated hypotheses.

Experimental design

The work represents original primary research that fits within the journal’s aims and scope. The research question is well defined and addresses a meaningful gap in the field, namely the limitations of single-scale mean field reinforcement learning in large-scale multi-agent systems. The experimental methodology is rigorous and described in sufficient detail to allow replication. Ethical and technical standards are adequately met.

Validity of the findings

The results are robust, statistically sound, and supported by clear experimental evidence across multiple benchmark tasks. All relevant data are presented, and the statistical analysis confirms the significance of the findings. The conclusions are logically derived from the results and remain well aligned with the initial research questions, without overstating the implications.

Additional comments

In its present form, the manuscript is ready for publication. It makes a clear and valuable contribution to the literature on multi-agent reinforcement learning, offering both theoretical novelty and validated experimental results.